# Adeno-Associated Virus (AAV)-Delivered Exosomal TAT and BiTE Molecule CD4-αCD3 Facilitate the Elimination of CD4 T Cells Harboring Latent HIV-1

**DOI:** 10.3390/microorganisms12081707

**Published:** 2024-08-18

**Authors:** Xiaoli Tang, Huafei Lu, Patrick M. Tarwater, David L. Silverberg, Christoph Schorl, Bharat Ramratnam

**Affiliations:** 1Division of Infectious Diseases, Department of Medicine, Warren Alpert Medical School, Brown University, Providence, RI 02903, USA; xiaoli_tang@brown.edu (X.T.); huafei.lu@lifespan.org (H.L.); 2Department of Epidemiology and Biostatistics, Texas A&M School of Public Health, College Station, TX 77843, USA; tarwater@tamu.edu; 3Department of Pathology & Laboratory Medicine, Brown University, Providence, RI 02906, USA; david_silverberg@brown.edu; 4The Brown University Genomics Core, Providence, RI 02906, USA; christoph_schorl@brown.edu; 5Department of Molecular Biology, Cell Biology and Biochemistry, Brown University, Providence, RI 02906, USA; 6COBRE Center for Cancer Research Development, Proteomics Core Facility, Rhode Island Hospital, Providence, RI 02903, USA; 7Clinical Research Center of Lifespan, Providence, RI 02903, USA

**Keywords:** HIV-1 latency, latency reversal agent, exosomal Tat, CD4-αCD3, adeno-associated virus (AAV)

## Abstract

Combinatorial antiretroviral therapy (cART) has transformed HIV infection from a death sentence to a controllable chronic disease, but cannot eliminate the virus. Latent HIV-1 reservoirs are the major obstacles to cure HIV-1 infection. Previously, we engineered exosomal Tat (Exo-Tat) to reactivate latent HIV-1 from the reservoir of resting CD4+ T cells. Here, we present an HIV-1 eradication platform, which uses our previously described Exo-Tat to activate latent virus from resting CD4+ T cells guided by the specific binding domain of CD4 in interleukin 16 (IL16), attached to the N-terminus of exosome surface protein lysosome-associated membrane protein 2 variant B (Lamp2B). Cells with HIV-1 surface protein gp120 expressed on the cell membranes are then targeted for immune cytolysis by a BiTE molecule CD4-αCD3, which colocalizes the gp120 surface protein of HIV-1 and the CD3 of cytotoxic T lymphocytes. Using primary blood cells obtained from antiretroviral treated individuals, we find that this combined approach led to a significant reduction in replication-competent HIV-1 in infected CD4+ T cells in a clonal in vitro cell system. Furthermore, adeno-associated virus serotype DJ (AAV-DJ) was used to deliver Exo-Tat, IL16lamp2b and CD4-αCD3 genes by constructing them in one AAV-DJ vector (the plasmid was named pEliminator). The coculture of T cells from HIV-1 patients with Huh-7 cells infected with AAV-Eliminator viruses led to the clearance of HIV-1 reservoir cells in the in vitro experiment, which could have implications for reducing the viral reservoir in vivo, indicating that Eliminator AAV viruses have the potential to be developed into therapeutic biologics to cure HIV-1 infection.

## 1. Introduction

Initiation of antiretroviral therapy (ART) decreases HIV-1 replication with the prompt reduction in plasma virus levels to below the detection limit of currently used commercial assays [1]. A small pool of infected lymphocytes survives initial infection, persists during therapy, and can rekindle viral replication upon stopping therapy [2,3]. The eradication of this latent viral reservoir remains a clinical and scientific challenge [4].

Five individuals have been cured of HIV-1 infection. In all cases, an otherwise fatal hematologic malignancy was treated by curative allogeneic bone marrow transplantation (BMT) with cells harvested from donors with a mutated HIV-1 CCR5 co-receptor gene (CCR5Δ32/Δ32) [5,6]. BMT is extremely expensive and is often accompanied by major complications, such as infection and graft-versus-host disease. While these cases prove the principle that HIV-1 can indeed be eradicated, the field continues to search for cure strategies that are feasible in resource-poor settings, which currently harbor the majority of infected individuals.

Several experimental strategies have emerged to eradicate HIV-1, including genetic, pharmacologic, and immune based therapies. One strategy, “Shock and Kill”, began with the unsuccessful trial of global immune-stimulation with OKT3 through ART, which led to toxic sepsis-like syndrome but no significant decrease in total body HIV-1 burden [7,8]. Since then, strategies have become more refined, with less off-target effects. Several small molecules such as disulfiram and SAHA can reverse latency in vitro, but this has not translated into reproducible reduction in the latent reservoir in human trials [9,10,11]. Other molecules such as mimetics of the second-mitochondrial-derived activator of caspases (SMAC) have recently been shown to potently reverse HIV-1 latency in humanized murine models of HIV-1 infection [12]. While several pharmacologic compounds activate the latent virus in vivo, few biologics have been identified that kill cells with a reactivated virus [13,14]. These molecules targeted certain signal pathways of host cells, but did not specifically aim at the virus itself. We have previously engineered exosomal Tat (Exo-Tat), which reactivates latent HIV-1 efficiently from primary CD4+ T cells from people with HIV (PWH) [15]. Here, we demonstrate that the addition of a retargeting protein with affinity to both gp120 and CD3 leads to the death of HIV-1-infected cells. We describe the generation and validation of this immunologic agent and the potential use of a viral platform to deliver Exo-Tat, IL16lamp2b and CD4-αCD3 to potentially eliminate HIV-1 reservoirs in primary lymphoid cells from PWH. Since AAV is nonpathogenic and several AAV-based biological therapeutics were approved by FDA for clinical applications, we intend to deliver the target genes using a single AAV vector to PWH to specifically reactivate latent HIV-1 to eliminate the reservoir cells via the immune system.

## 2. Materials and Methods

### 2.1. Cell Culture and Transfection

HEK293T cells (ATCC) and Huh-7 cells (Japanese Collection of Research Bioresources) were cultured in Dulbecco’s modified Eagle’s medium (Life Technologies) with 10% fetal bovine serum (FBS) (Thermo Scientific), 2 mM l-glutamine and non-essential amino acids (Life Technologies). MOLT-4/CCR5 cells (CCR5-expressing MOLT-4 cells from NIH Aids Reagents), U1 cells (NIH Aids Reagents), primary human peripheral blood mononuclear cells (PBMCs) and human T cells were cultured in RPMI Medium 1640 (Life Technologies) with 10% FBS. AAV-Eliminator-infected Huh-7 cells were cocultured with T cells in RPMI Medium 1640. Adherent cells were trypsinized and re-seeded in culture plates 1 day before transfection or treatment. HEK293T cells were transfected with Lipofectamine when cell confluency was ~70%. MOLT-4/CCR5 cells were transfected with Lipofectamine LTX Plus Reagent (Life Technologies).

### 2.2. Exosome Preparation

The cDNA fragments encoding HIV-1 Tat protein with a myc nuclear localization signal fused to its C-terminus were subcloned into XPack CMV-XP-MCS-EF1-Puro Cloning Lentivector (System Biosciences) to generate pEXO-Tat. The cDNA fragment encoding the C-terminal domain (Thr102 to Ser121) of interleukin 16 fused to the N-terminus of lysosome-associated membrane protein 2 variant b (lamp2b) was cloned into pCDH-EF1-MCS-T2A-Puro (System Biosciences) to generate pIL16lamp2b. A lentiviral packaging plasmid pPACKH1 (System Biosciences) was cotransfected into HEK293T cells with pIL16lam2b or pEXO-Tat at the ratio 2:1 to generate IL16lamp2b or EXO^CD4^-Tat lentiviruses. Lentiviruses were concentrated with PEG-it^TM^ Virus Precipitation Solution (System Biosciences) and used to transduce HEK293T cells to generate stable cells under the pressure of puromycin selection [15]. Exo-Tat stable cells were cultured in Dulbecco’s modified Eagle’s medium (Life Technologies) with 10% fetal bovine serum (FBS) (Thermo Scientific), 2 mM l-glutamine and non-essential amino acids (Life Technologies) and 5 µg/mL puromycin. The supernatants of stable cells were collected for the isolation of exosomes using a differential ultracentrifugation method. The number and size distribution of exosomes were determined on a NanoSight NS500 (Malvern Instruments, Malvern, UK) with a Syringe Pump. For treatment, 50 µL of exosomes (1.8 × 10^9^ exosomes or 46.8 µg total protein) or 50 ng/mL PMA plus 1 µM ionomycin was added to 450 µL culture medium of T cells as previously described [15]. The control exosomes were prepared using the same procedure, but replacing the supernatants of Exo-Tat stable cells with the those of HEK293T cells.

### 2.3. Molecular Cloning

To allow for the convenient detection of the fusion protein, an HA-tag was added to the C-terminus of each target protein. The cDNA fragment encoding CD4-αCD3 (domains 1 and 2 of CD4 were fused to an anti-CD3 single-chain antibody fragment with an HA-tag on the C-terminus) was subcloned into an expression vector pAAV-MCS (System Biosciences) between enzyme sites ClaI and BglII. The original cDNA fragments were synthesized at Integrated DNA Technologies (IDT). The generated construct was named pAAV-CD4-αCD3. The cDNA sequences encoding Exo-TatHA-F2A-IL16lamp2bHA-F2A-CD4-αCD3 were subcloned into pAAV-MCS between enzyme sites SalI and BglII. The generated construct was named pEliminator. All constructs were sequenced at Yale Keck Biotechnology Resource Laboratory. All primers used for molecular cloning were listed in Appendix A.

### 2.4. Generation of CD4-αCD3

The expression construct pAAV-CD4-αCD3 (30 µg) was transfected into 30 × 10^6^ MOLT-4/CCR5 cells (1 × 10^6^ cells/mL) using lipofectamine LTX plus reagent. Twenty-four hours post-transfection, the culture medium was changed to fresh medium. The supernatant was collected on day 2 and day 3 after transfection. The combined supernatant (60 mL) was cleared by centrifugation (3000 rpm for 10 min) and used as CD4-αCD3 medium. The expression level of small protein molecule CD4-αCD3 in the combined supernatant was measured by immunoprecipitation–Western blot method. A control was prepared under the same conditions but using the backbone vector pAAV-MCS instead of pAAV-CD4-αCD3 for transfection. Since T cells and MOLT-4/CCR5 cells use the same culture medium, we did not purify CD4-αCD3 from the supernatant of transfected MOLT-4/CCR5 cells but added the CD4-αCD3 medium to T cells culture at a 1:1 ratio.

### 2.5. Immunoprecipitation

One milliliter of combined supernatant was used for immunoprecipitating fused protein CD4-αCD3 with anti-HA rabbit monoclonal antibody sepharose beads (Cell Signaling). The combined supernatant and antibody were rotated at 4 °C overnight. The precipitates were washed three times using Pierce IP lysis buffer (Thermo Scientific) and eluted in 2× NuPAGE LDS Samples Buffer (Life Technologies) for Western blot.

### 2.6. Western Blot

Immunoprecipitated protein or cell lysate (10–20 μg total protein) was mixed with NuPAGE LDS Samples Buffer and separated by 4–12% NuPAGE Novex 4–12% Bis-Tris gel electrophoresis and electroblotted to nitrocellulose membrane (Bio-Rad). Blotted membranes were probed with their respective primary antibodies, rotating at 4 °C overnight. Membranes were washed three times in TBST buffer and probed with secondary antibody (680 goat anti-rabbit IgG or IRDye800-conjugated Affinity Purified Anti-Mouse IgG, respectively) at room temperature for 1 h. Membranes were then washed three times in TBST buffer and direct infrared fluorescence detection was performed with a Licor Odyssey Infrared Imaging System [16].

### 2.7. HIV-1 p24 Elisa Assay

U1 cells were cocultured with purified human CTLs in the control or CD4-αCD3 medium for 24 h. The cells and culture medium were separated by centrifugation. The cell pellet and supernatant from each well were used for determining the p24 levels using an Alliance HIV-1 ELISA Kit (PerkinElmer Inc.) following the manufacturer’s instructions. The analytical sensitivity of the kit is 17.1 pg/mL.

### 2.8. Study Subjects

In this study, we used the PBMCs of 11 people living with HIV-1 (PLWH) from a previous clinical study which recruited 223 PLWH participants from 12 September 2017 to 15 May 2018 to investigate the effects of recreational drug use on HIV-1 infection at The Miriam Hospital based on the criteria of suppressive antiretroviral therapy (ART) and undetectable plasma HIV-1 RNA levels (<50 copies per mL) for a minimum of 12 months. The study was approved by Lifespan Institutional Review Board. All research participants enrolled in the study provided written, informed consent prior to inclusion in this study.

### 2.9. Study Approval

The study was approved by Lifespan Institutional Review Board. The study IRB number is 2100-17. All experiments were performed in accordance with relevant guidelines and regulations.

### 2.10. Isolation of Human T Cells

Peripheral blood mononuclear cells (PBMCs) from whole blood of PWH were purified using density centrifugation on a Ficoll-Hypaque (GE Healthcare) gradient. Human T lymphocytes with high purity and viability were isolated from PBMCs by negative depletion using an Invitrogen Dynabeads Untouched Human T cells kit (Life Technologies) following the manufacturer’s instructions.

### 2.11. Isolation of Cytotoxic T Lymphocytes (CTLs)

PBMCs from whole blood of healthy donors were purified using density centrifugation on a Ficoll-Hypaque (GE Healthcare) gradient. CTLs were isolated from PBMCs using an EasySep™ Human CD8+ T Cell Isolation Kit (Stemcell Technologies) following the manufacturer’s instructions.

### 2.12. Measurement of Intracellular HIV-1 mRNA

Five million T cells were treated with control exosomes (Exo-C) or Tat exosomes (Exo-Tat), respectively, for 4 days in the presence of antivirals (1 μM tenofovir, 1 μM nevirapine, 1 μM emtricitabine). On day 5, T cells and supernatants were separated by centrifugation. The cells were washed with PBS. Half of the T cells were used to detect the intracellular HIV-1 mRNA level using RT-qPCR on the ***ViiA 7*** Real-Time PCR System (Thermo Fisher Scientific) as previously described [15,17]. Primers and probe (Integrated DNA Technologies) used for HIV-1 mRNA measurement were as previously described [18]: forward (5′→3′) CAGATGCTGCATATAAGCAGCTG (9501–9523), reverse (5′→3′) TTTTTTTTTTTTTTTTTTTTTTTTGAAGCAC (9629–poly A), probe (5′→3′) FAM-CCTGTACTGGGTCTCTCTGG-MGB (9531–9550). The other half of the T cells was used for the HIVE assay.

### 2.13. HIV Elimination (HIVE) Assay

The principle of the experiment is to activate latent HIV-1 with an LRA in the presence of antivirals (1 μM tenofovir, 1 μM nevirapine, 1 μM emtricitabine) to let the cells with gp120 expressed on the surface be eliminated by cytotoxic T cells (CTLs) in the activation/elimination part. The remaining reservoir cells are stimulated with PHA and IL-2 for 14 days to maximally activate latent HIV-1 in the HIVE assay part. The p24 level in the final coculture medium indicates the uneliminated reservoir. HIVE assay was performed following a previous study with some modifications [14]. Briefly, 5 million human T cells were cultured in control or CD4-αCD3 medium supplemented with penicillin-streptomycin, L-glutamine, 0.1 nM IL-7, 1 μM tenofovir, 1 μM nevirapine, 1 μM emtricitabine and treated with control exosomes (Exo-C), Tat exosomes (Exo-Tat), DMSO solvent control or PMA/I, respectively, for 4 days in the presence of antivirals. The cells and supernatants were separated by centrifugation. Half of the T cells were used to check activation by measuring the intracellular HIV-1 mRNA level. The other half of the T cells were cocultured with 2 million MOLT-4/CCR5 cells and irradiated PBMCs in a HIVE assay medium containing 2.5 µg/mL of PHA and 60 U/mL of IL-2 for a further 14 days to maximally activate T cells. The culture medium was changed every 3 days. The final culture supernatants were used for measuring p24 levels using Simoa technology (Quanterix) [15]. The analytical sensitivity of the Simoa technology is 0.0074 pg/mL.

### 2.14. Production of AAV Viruses

Control vector pAAV-GFP or expression vector pEliminator was cotransfected with pHelper and pAAV-DJ at a molar ratio of 1:1:1 into HEK293T cells using Lipofectamine 2000 Reagent (Invitrogen). Seventy-two hours post-transfection, the cells were harvested for purifying AAV viruses. Briefly, the cells were trypsinized and suspended in culture medium. The cell suspension was centrifuged at 1750× *g* at 4 °C for 10 min. The supernatant was discarded, and the cell pellet was used for AAV preparation using TAKARAAAVpro Purification Kit Maxi (All Serotypes) following the instructions of the manufacturer. Viral titration was performed using QuickTiter AAV Quantitation Kit (Cell Biolabs).

### 2.15. Infection of Balb/cJ Mice

The experimental protocol was approved by the Lifespan Animal Care and Use Committee. The approval number is 503720. All experimental procedures were conducted in accordance with guidelines for the ethical treatment of animals. Balb/cj mice (3 months old) were purchased from Jackson Laboratory. AAV-GFP (Control) or AAV-Eliminator (Eliminator) viruses were injected into the mice via the tail veins (n = 3, 1 × 10^12^ GC/mouse in 200 μL PBS) as previously described [19]. The mice were kept in a BL2 animal facility at Rhode Island Hospital and taken care of by a certified technician. The mice were injected only once and euthanized on day 31 by overdose isoflurane. Blood was taken by cardiac puncture before death. Brain, heart, liver, spleen, kidney, lung, lymph nodes (LN) and muscle were taken after death.

### 2.16. Immunohistochemistry (IHC) and Hematoxylin and Eosin (H&E) Staining

After the mice were euthanized with overdose isoflurane, one piece (2 mm thick) of brain, heart, liver, spleen, kidney, lung, lymph node was fixed in 4% paraformaldehyde solution overnight and then kept in 70% ethanol. IHC and H&E staining was performed at The Molecular Pathology Core at Brown University using well-established standard procedures. The primary antibody for IHC staining was HA-Tag (6E2) Mouse mAb (Cell Signaling). The secondary antibody used for IHC staining was Horse anti-Mouse IgG ImmPRESS (TM) Secondary Antibody [HRP Polymer] from Vectorlabs.

### 2.17. Intact Proviral DNA Assay (IPDA)

One million Huh-7 cells were seeded in each well of a 6-well plate in DMEM medium and infected with control AAV or AAV-Eliminator at 1 × 10^4^ multiplicity of infection (MOI). Forty-eight hours post-infection, the Huh-7 cells were washed with PBS and cocultured in RPMI Medium 1640 containing antivirals (1 μM tenofovir, 1 μM nevirapine, 1 μM emtricitabine), with 4 million T cells isolated from the blood of PWH. Four days post coculture, the T cells were collected without disturbing the attached Huh-7 cells. Total DNA was purified from the T cells using a GeneJET Genomic DNA Purification Kit (Thermo Scientific) and used for IPDA analysis using primers and probes as previously described [20]. Droplet digital PCR was performed on the Bio-Rad QX200 Digital Droplet PCR system using the appropriate manufacturer-supplied consumables. For DNA shearing and copy number reference reactions, 3 µL out of the 120 µL of genomic DNA were analyzed in each reaction well. Three replicate wells were performed for each patient sample. After correction for DNA shearing, results are expressed as HIV-1 proviral copies in 3 replicate wells for each patient sample.

### 2.18. Statistical Analysis

Quantitative data were analyzed by parametric tests (paired *t*-test and two independent sample *t*-test with unequal variances), analogous nonparametric tests (sign rank and rank sum) and unpaired Student’s *t* test. The *p* values did not differ inferentially with the statistical test used. Data are expressed as *mean ± standard error of mean*. A *p* value < 0.05 indicates statistical significance.

## 3. Results

### 3.1. Cytotoxic T Lymphocytes (CTLs) from Healthy Donor Blood Eliminate HIV-1-Infected Cells

CTLs are vital defense cells against the infection of host cells by viruses and other pathogens. They are also critical components of antitumor immunity in the body. We began our experiments by testing whether CTLs from healthy donor blood eliminate HIV-1-infected cells. U1 cells are derived from U937 cells that have been chronically infected with HIV-1. The U937 cell line is a pro-monocyte from a pleural effusion of a two-year-old Caucasian male with diffuse histiocytic lymphoma [21]. The cells show a constitutive expression of HIV-1 and secrete virions into the culture medium. Therefore, U1 cells can be used as an HIV-1-infected cell model system. When U1 cells die, p24 molecules are released into the culture medium. An intracellular p24 decrease or extracellular p24 increase can be used as an indirect cell death indicator. 1.25 × 10^5^ U1 cells were cocultured overnight with 5 million CTLs purified from healthy donor blood. As expected, CTLs killed U1 cells, leading to a dramatically increase in the p24 level in the supernatant (Figure 1A) and a significantly decrease in the intracellular p24 level (Figure 1B).

### 3.2. Autologous CTLs Fail to Effectively Eliminate CD4+ T Cells with Latent HIV-1 Reactivated by Exo-Tat

We expected Exo-Tat would reactivate latent HIV-1 virus from CD4 T cells, and those reservoir cells with HIV-1 reactivated would be eliminated by autologous CTLs. For these experiments, we used highly purified populations of T lymphocytes. These T cells were isolated from PWH (patient IDs: Lu103, Lu109, Lu206, Lu204 and Lu202) on long-term, suppressive ART (5–12 years). Exo-Tat was used to activate the latent virus, as described previously [15]. Five million purified T cells were treated with Exo-C or Exo-Tat exosomes in the presence of antivirals for 4 days. Half of the T cells were used to check HIV-1 activation by RT-qPCR. The treatment of T cells from HIV-1 patients with Exo-Tat exosomes for 4 days led to the increase in intracellular HIV-1 mRNA in all five samples, indicating the reactivation of latent HIV-1 by Exo-Tat but not by Exo-C (Figure 2A). The other half of the T cells treated with Exo-C or Exo-Tat exosomes were then subjected to an HIV Elimination (HIVE) assay as described previously, with some modifications [14]. In the HIVE assay, the remaining reservoir cells are maximally stimulated with Phytohemagglutinin (PHA) and Interleukin-2 (IL-2) to activate latent HIV-1. The p24 antigen levels in final culture supernatants were measured using Simoa Technology (Figure 2B). To our surprise, the reactivation of latent HIV-1 by Exo-Tat did not appreciably lead to the clearance of HIV-1 reservoir cells by autologous CTLs, indicating that autologous *immune incompetence* needs to be overcome by a specific immune enhancer such as a bispecific T cell engager (BiTE).

### 3.3. Generation of a BiTE Molecule CD4-αCD3

The observation that autologous CTLs do not effectively kill CD4+ T lymphocytes with reactivated HIV-1 has been made previously [13,22]. We reasoned that biologics capable of approximating both CD8+ T cells and HIV-1 Env-expressing cells would improve the killing activity of latency-reversing agents (LRAs) such as Exo-Tat. To facilitate the killing, we used a BiTE molecule CD4-αCD3 which was reported in previous studies [23,24,25]. This molecule specifically recognizes both gp120 and cytotoxic CD8+ T cells by fusing domain 1 and domain 2 of CD4 with the single-chain fragment variable (scFv) of an anti-CD3 antibody via a five amino acid linker (Figure 3A). We verified the expression of the generated plasmid, pAAV-CD4-αCD3, in HEK293T cells (Figure 3B). After pAAV-CD4-αCD3 was transfected into MOLT-4/CCR5 cells, CD4-αCD3 protein could be detected in the culture medium by immunoprecipitation–Western blot indicating that this protein could be secreted into culture medium (Figure 3C).

### 3.4. CD4-αCD3 Mediates Killing of Cells Stably Expressing HIV-1

Our initial experiment showed that CTLs from healthy donor blood eliminated U1 cells, a pro-monocytic cell line engineered to harbor integrated HIV-1. All HIV-1 latent cell lines, including U1 cells, have noticeable levels of baseline viral expression that increase after treatment with agents that activate the HIV-1 LTR. We wondered if CD4-αCD3 facilitates the killing of U1 cells by CTLs. U1 cells were co-cultured with cytotoxic T cells (CTLs) in the presence of control medium or CD4-αCD3. Co-culture of U1 cells and CTLs in the presence of CD4-αCD3 resulted in the elevation of the p24 level in the medium (Figure 4A) and the reduction in the intracellular p24 levels of U1 cells (Figure 4B), indicating that CD4-αCD3 facilitates the killing of U1 cells by CTLs from healthy donor blood.

### 3.5. CD4-αCD3 Mediates Killing of Chronically HIV-1-Infected Primary CD4+ T Lymphocytes

While these results in U1 cells were encouraging, the HIV-1-infected cell number in U1 cells is different from that in primary lymphocytes, with almost 100% of U1 cells infected by HIV-1 harboring the replication-competent latent virus as opposed to <1% of primary CD4+ T lymphocytes harboring the replication-competent latent virus. We first used PMA/I as a positive LRA to activate resting CD4+ T lymphocytes in the presence of antivirals and tested the efficacy of CD4-αCD3 in primary CD4+ T cells from HIV-1-infected and ART-treated patients (patient IDs: Lu107, Lu203, Lu215 and Lu218). As seen in Figure 5A, the HIV-1-infected reservoir was decreased by 60% in the presence of CD4-αCD3 compared to experiments employing control conditions. We then activated latent HIV-1 from the CD4+ T cells of five PWH (patient IDs: Lu103, Lu109, Lu206, Lu204 and Lu202) with Exo-Tat exosomes. In the presence of antivirals, five million purified T cells were treated with Exo-C/control medium or Exo-Tat/CD4-αCD3 medium for 4 days. Half of the T cells treated with Exo-C or Exo-Tat exosomes were then subjected to an HIV Elimination (HIVE) assay as mentioned above. Compared to control experiments employing control conditions (Exo-C/control medium without CD4-αCD3), Exo-Tat/CD4-αCD3 reduced the HIV-1-infected cell number by 90%, as indicated by the p24 level in the supernatant measured using Simoa Technology (Figure 5B), indicating that the combination of exosomal Tat with BiTE molecule CD4-αCD3 can eliminate HIV-1-expressing cells. Half of the control, Exo-Tat or PMA/I treated T cells were used to check HIV-1 activation by RT-qPCR. As shown in Figure 5C, PMA/I or Exo-Tat reactivated latent HIV-1, increasing mRNA expression.

### 3.6. Construction of an AAV Vector Encoding Exo-Tat, IL16lamp2b and CD4-αCD3

The above promising in vitro data were achieved using raw materials, Exo-Tat exosomes and CD4-αCD3. Exo-Tat exosomes were prepared from the supernatants of HEK293 cells stably expressing Exo-Tat and IL16lamp2b. The CD4-αCD3 reagent was the supernatants of MOLT-4/CCR5 cells transfected with a CD4-αCD3 expression vector. The raw supernatant containing CD4-αCD3 was used as the BiTE reagent without further purification. These raw materials are very hard, if not impossible, to quantitate. To facilitate delivery and increase reproducibility, we decided to deliver the genes encoding Exo-Tat, IL16lamp2b and CD4-αCD3 in one carrier. Since adeno-associated virus (AAV) is non-pathogenic and some AAV based gene therapies are approved by the FDA, we decided to use AAV as the carrier to deliver the target genes. We constructed an AAV-DJ vector encoding Exo-Tat, IL16lamp2b and CD4-αCD3 by linking them with self-cleaving F2A peptides [26]. The newly constructed plasmid was named pEliminator. AAV-Eliminator viruses were produced by cotransfecting pEliminator with pHelper and pAAV-DJ at a molar ratio of 1:1:1 into HEK293T cells. The infection of MOLT-4/CCR5 cells with AAV-Eliminator viruses results in the protein expression of Exo-Tat, IL16lamp2b and CD4-αCD3 in MOLT-4/CCR5 cells (Figure 6A). Exo-Tat and IL16lamp2b can be detected in the exosomes purified from the supernatant of infected MOLT-4/CCR5 cells (Figure 6B). CD4-αCD3 is secreted into the culture medium and can be detected in the exosome-free supernatant (Figure 6C). We have not observed any toxic effects of AAV-Eliminator on cultured cells. We wondered if the target genes can be delivered in vivo by injection of AAV-Eliminator viruses into Balb/cJ mice via the tail veins. Our data indicated that AAV-Eliminator can deliver target genes to various tissues as shown by Western blot data (Figure 7A) and immunohistochemistry (IHC) results (Figure 7B). These gene products do not change the cellular and tissue structure of infected cells, as manifested by H&E staining (Figure 7C). An HA-tag was added to the C-terminus of Exo-Tat, IL16lamp2b and CD4-αCD3 for the convenience of detecting protein expression. After a one-time injection of AAV-Eliminator into Balb/cJ mice for 31 days, HA-tagged proteins could be detected in the blood at concentrations of 18.6 pg/mL using a Human HA Tag ELISA kit (MyBioSource) with an analytical sensitivity of 1.0 pg/mL.

### 3.7. AAV-Eliminator Reduces Latent HIV-1 Reservoir in T Cells Isolated from PWH

AAV-Eliminator is based on an AAV-DJ vector which efficiently infects a plethora of cell types [27]. Our notion is that AAV-Eliminator will infect various cells in the body and transform the infected cells into factories to produce CD4-targeting Exo-Tat exosomes and CD4-αCD3 molecules. Exo-Tat exosomes will specifically target CD4+ reservoir cells to reactivate latent HIV-1. CD4-αCD3 molecules will mediate the killing of HIV-1 gp120-expressing reservoir cells. To mimic this situation, Huh-7 cells were infected with control AAV-GFP (Control) or AAV-Eliminator viruses and cocultured with T cells isolated from PWH PBMCs in the presence of antivirals for 4 days. Four days later, T cells were collected and used for total DNA preparation. The intact HIV-1 proviruses were determined by intact proviral DNA assay (IPDA) using a droplet digital polymerase chain reaction (ddPCR) system (Bio-Rad). IPDA is a novel quantitative approach for measuring the reservoir of latent HIV-1 proviruses [20]. As a proof of concept, we confirmed, in 3 out of 6 patient samples (patient IDs: Lu210, Lu107 and Lu223), that the latent HIV-1 reservoirs were reduced when T cells from PWH were cocultured with AAV-Eliminator-infected Huh-7 cells (Figure 8).

## 4. Discussion

HIV-1 eradication platforms must effectively identify and kill infected cells harboring a replication-competent virus. Multiple challenges abound [28]. For example, no specific host marker currently identifies infected cells either peripherally or in central nervous system tissue. While most proviruses are replication-incompetent, the ~1% that are replication-competent appear distributed in several lymphoid and myeloid subpopulations. In the setting of prolonged, suppressive ART, lymphoid viral reservoirs are classically defined as harboring latent forms of a virus. Myeloid reservoirs; however, are less defined but thought to persist due to the inherent resistance of macrophages to HIV-1-induced cytopathicity [29]. Thus, it is no surprise that a HIV-1 cure has only been accomplished with allogeneic hematologic cell transplantation and the global cellular repopulation that ensues, an approach untenable for the majority of infected individuals.

Some immunotherapeutic strategies to eliminate latent HIV-1 reservoirs appear to be very promising [7,8,30]. One strategy called “shock and kill” sheds light on the cure of HIV-1 infection. The first key step of this strategy is to reactivate latent HIV-1 from its reservoirs [7,8]. Previously, we tested some latency reversal agents (LRAs) and found that HIV-1 Tat was the most potent LRA [15]. Tat is expressed by HIV-1 early in the life cycle of the virus. Tat binds to the transactivation response element (TAR) to activate the long terminal repeat (LTR) and is critical for viral replication and progression to disease [31]. In this study, we used an HIV-1 Tat plasmid from Addgene (Plasmid #14654), which is contributed by the Matija Peterlin Lab [32]. This plasmid encodes a Tat sequence of HIV-1 subtype B, the most prevalent HIV-1 strain which accounts for approximately 11% of all cases of HIV infection globally [33]. Tat variants from different HIV-1 strains may function differentially, but all subtype LTRs responds equally well to the Tat trans activator protein of subtype B [34]. The Tat protein of subtype B can be used as a universal LRA to activate all HIV-1 strains.

Here, we experimentally validate a two-step recombinant viral platform designed to reverse latency and eliminate reservoir cells harboring replication-competent HIV-1. In the first step, we use our previously described CD4+ receptor, targeting exosomes harboring HIV-1 Tat, to activate the latent virus [15]. We used purified T cells instead of separating CD4+ T cells and CD8+ T cells because total T cells reflect the real ratio of CD4/CD8 in the body. To our surprise, Tat-mediated activation alone did not lead to the significant killing of infected cells by autologous CTLs. However, when this activation step was combined with a bispecific molecule CD4-αCD3, we found that on average >90% of HIV-1-infected T cells reactivated by Exo-Tat are killed by autologous CTLs.

One concern is that CD4-αCD3 may lead to the HIV-1 infection of CD4-CD8+ T cells [24]. Under the cover of antiretrovirals, this possibility is decreased. Previous studies using BiTEs or DARTs usually purified the BiTE or DART molecules from the supernatants and added the purified molecules to culture media of target and effector cells [24,35,36]. As a proof of concept, we transfected an expression vector pAAV-CD4-αCD3 into lymphocyte-derived MOLT-4/CCR5 cells to generate CD4-αCD3. Since the same culture media were used for CD4-αCD3 preparation and the functional study, further purification of CD4-αCD3 from the supernatants was omitted. A control supernatant was prepared under the exact same conditions except replacing pAAV-CD4-αCD3 with an empty backbone vector pAAV-MCS, into which CD4-αCD3 was subcloned. We determined the level of CD4-aCD3-HA in the supernatant by Western blot and confirmed its amino acid sequence using LC-MS/MS Mass Spectrometry analysis.

When purified Exo-Tat exosomes and CD4-αCD3 molecules are delivered into the body, the maintenance of the constant effective concentrations of these agents is a big concern. We reasoned that delivering these agents in one single carrier would provide a cleaner delivery system and result in constant LRA Exo-Tat and BiTE molecule CD4-αCD3 production over a prolonged period. Since AAV can exist in different types of cells for a long time and is the only non-pathogenic virus with prospective applications in delivering therapeutic genes to various patients [37,38], we intend to use AAV to deliver the target genes into the body and transform the infected cells into manufactories to produce Exo-Tat exosomes and CD4-αCD3. Our previous study showed that AAV-delivered Exo-Tat is nontoxic to mice [19]. We have constructed an AAV vector encoding the cDNA sequences of Exo-Tat, IL16lamp2b and CD4-αCD3 (pEliminator). AAV-Eliminator viruses were produced in the laboratory by cotransfecting pEliminator with pHelper and pAAV-DJ into HEK293T cells. After intravenous injection of purified AAV-Eliminator viruses into Balb/cJ mice, Exo-Tat, IL16lamp2b and CD4-αCD3 can be detected in various tissues and blood, consistent with previous studies [19,39]. As proof of concept, the function of AAV-Eliminator viruses to clear HIV-1 reservoirs is validated in 3 out of 6 PWH samples (Figure 8). IPDA amplicon signal failure may be due to the polymorphism of HIV-1 proviral DNA [40] or not enough CD4+ T cells used initially. The primers and probes used in the IPDA are based on HIV-1 clade B alignments [20]. AAV-Eliminator viruses will be used in future in vivo validation studies in humanized mice. Exo-Tat, IL16lamp2b and CD4-αCD3 were effectively packaged in an AAV vector, raising the possibility of achieving HIV-1 reservoir targeting with an injectable biologic that may persistently produce eradication reagents in vivo. These data provide a scientific basis for further in vivo studies in humanized mice models. Whether the exosomes can be specifically targeted to CD4+ cells in vivo is another concern. Both CD4 and CD9 are the receptors of IL-16. IL-16 binds to the D4 domain of CD4 using its C-terminal domain with minimal peptide sequence RRKS (Arg106–Ser109). We cloned the C-terminal 20 aa (Thr102-Ser121) of IL-16 into Lamp2b to create a fusion protein IL16lamp2b for targeting Exo-Tat exosomes to CD4+ T cells. IL16lamp2b specifically binds to CD4 but not CD9 (Appendix A). The alignment of CD4 and CD9 amino acid sequences reveals no domain in CD9 that is similar to the D4 domain of CD4, indicating a domain other than the C-terminal 20 aa in IL-16 binds to CD9.

## Figures and Tables

**Figure 1 microorganisms-12-01707-f001:**
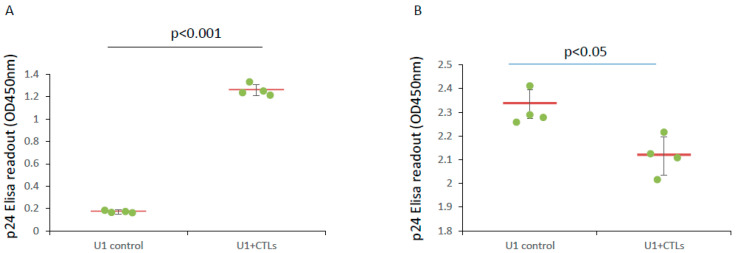
Cytotoxic T lymphocytes (CTLs) from healthy donor blood clear U1 cells. U1 cells were cocultured with CTLs from healthy donor blood at target:effector = 1:40 in RPMI1640 medium containing 10% fetal bovine serum and 100 units penicillin/100 μg streptomycin/mL. Twenty-four hours later, cells and supernatant were separated by centrifugation. Cell pellets were lysed with Pierce IP Lysis Buffer. The p24 levels in supernatants and cell lysates were measured using an Alliance HIV-1 ELISA Kit (PerkinElmer Inc.) following the manufacturer’s instructions. The OD450nm readouts are shown as relative p24 levels. (**A**) CTLs kill U1 cells, leading to the release of p24 into culture medium. (**B**) CTLs kill U1 cells, leading to cell loss and total intracellular p24 level decrease.

**Figure 2 microorganisms-12-01707-f002:**
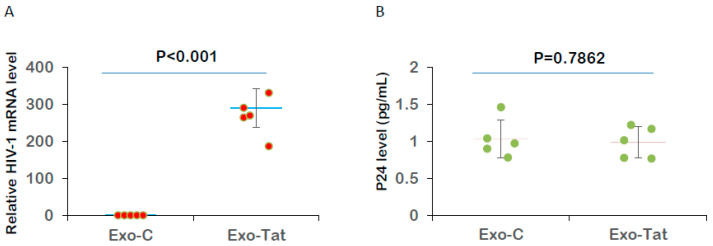
Autologous cytotoxic T cells fail to effectively eliminate CD4+ T cells harboring reactivated HIV-1. Five million T cells from the blood of cART-treated HIV-1-infected individuals were cultured in RPMI 1640 medium supplemented with penicillin-streptomycin, L-glutamine, 0.1 nM IL-7, 1 μM tenofovir, 1 μM nevirapine, 1 μM emtricitabine and treated with control exosomes (Exo-C) or Exo-Tat exosomes for 4 days. On day 5, half of the T cells were used to check the activation of latent HIV-1 by measuring the intracellular HIV-1 mRNA level using RT-qPCR (**A**). Another half of the T cells were cocultured with MOLT-4/CCR5 cells and irradiated PBMCs in an HIVE assay medium containing 2.5 μg/mL of Phytohemagglutinin (PHA) and 60 U/mL of IL-2 for an additional 14 days. The culture medium was changed every 3 days. The final culture supernatants were used for measuring p24 using Simoa Technology with an analytical sensitivity of 0.0074 pg/mL (**B**).

**Figure 3 microorganisms-12-01707-f003:**
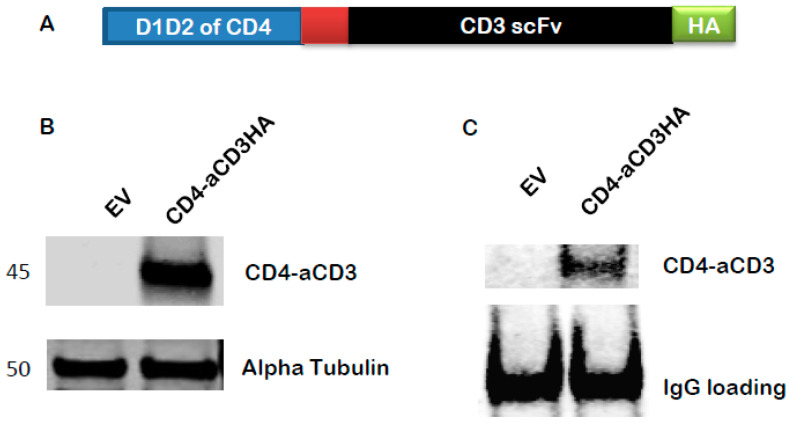
Construction and expression of CD4-αCD3. (**A**) Schematic structure of CD4-αCD3. The domains 1 and 2 of CD4 are fused to the single-chain variable fragment of anti-CD3 via a linker. An HA-tag is attached to the C-terminus of the fused protein for the convenience of measuring the expression level of the fusion protein. (**B**) Expression of CD4-αCD3 in HEK293T cells. An empty vector pAAV-MCS (EV) or expression vector pAAV-CD4-αCD3 was transfected into HEK293T cells, respectively. The protein expression level of CD4-αCD3 was determined by Western blot. (**C**) Secretion of CD4-αCD3 into culture medium of MOLT-4 cells. MOLT-4 cells were transfected with EV or pAAV-CD4-αCD3. Forty-eight hours post-transfection, the supernatants were collected and precipitated with anti-HA rabbit monoclonal antibody (sepharose beads conjugate). The precipitates were used for Western blot to measure levels of secreted CD4-αCD3.

**Figure 4 microorganisms-12-01707-f004:**
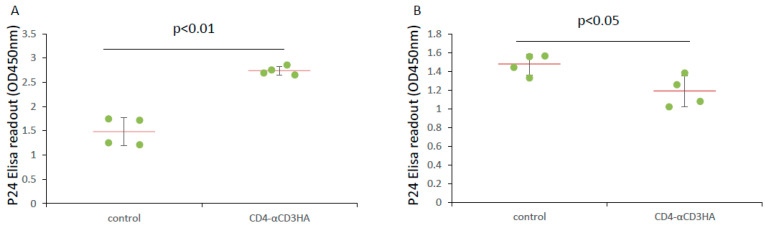
CD4αCD3 mediates the killing of U1 cells by cytotoxic T lymphocytes. U1 cells and CTLs from healthy donor blood were cocultured at target:effector = 1:40 in RPMI1640 medium plus control solution (control) or CD4αCD3 solution. Twenty-four hours later, cells and supernatant were separated by centrifugation. Cell pellets were lysed with Pierce IP Lysis Buffer. The p24 levels in supernatants and cell lysates were measured using an Alliance HIV-1 ELISA Kit (PerkinElmer Inc.) following the manufacturer’s instructions. (**A**) CD4αCD3 facilitates the killing of U1 cells by CTLs leading to further increase in extracellular p24 level. (**B**) CD4αCD3 facilitates the killing of U1 cells by CTLs leading to further cell loss and total intracellular p24 level decrease.

**Figure 5 microorganisms-12-01707-f005:**
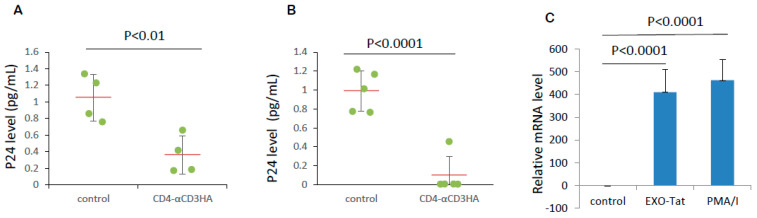
CD4-αCD3 mediates the elimination of HIV-1-infected resting CD4+ T cells ex vivo. (**A**) CD4-αCD3 mediates the elimination of CD4+ T cells with HIV-1 reactivated by PMA/I. Five million T cells from the blood of cART-treated HIV-1-infected individuals were cultured in control or CD4-αCD3 medium supplemented with penicillin-streptomycin, L-glutamine, 0.1 nM IL-7, 1 μM tenofovir, 1 μM nevirapine, 1 μM emtricitabine and treated with solvent control or PMA/I for 18 h. The cells were washed three times with regular RPMI 1640 culture medium to remove PMA/I. Half of the T cells were cocultured with 2 million MOLT-4/CCR5 cells and irradiated PBMCs in an HIVE assay medium containing 2.5 μg/mL of PHA and 60 U/mL of IL-2 for a further 14 days. The culture medium was changed every 3 days. The final culture supernatants were used for measuring p24 levels using Simoa technology. (N = 4, *p* < 0.01). (**B**) The combination of Exo-Tat and CD4-αCD3 eliminates latent HIV-1 reservoir ex vivo. The experimental procedure was the same as mentioned in Figure 5A, except that the T cells were treated with Exo-C exosomes or Exo-Tat exosomes instead of solvent control or PMA/I for 4 days. On day 5, half of the T cells were cocultured with 2 million MOLT-4/CCR5 cells and irradiated PBMCs in an HIVE assay medium containing 2.5 μg/mL of PHA and 60 U/mL of IL-2 for a further 14 days. The final culture supernatants were used for measuring p24 levels using Simoa technology(N = 5, *p* < 0.0001). (**C**) LRAs PMA/I or Exo-Tat reactivate latent HIV-1. T cells treated with or without PMA/I or Exo-Tat were used to detect intracellular HIV-1 mRNA level using RT-qPCR on theViiA 7Real-Time PCR System.

**Figure 6 microorganisms-12-01707-f006:**
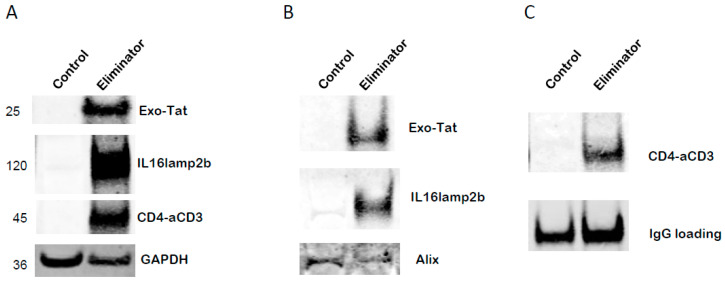
AAV delivered protein expression in vitro. Molt-4/CCR5 cells were infected with AAV-Eliminator at MOI = 1 × 10^4^. Three days post inoculation, cells and supernatant were separated by centrifugation. The cell pellet was used to prepare cell lysate using IP lysis buffer. Twenty μL of cell lysate were used for Western blot to determine the expression levels of HA-tagged proteins. The supernatant was used to purify exosomes. Twenty μL of exosomes were used for checking the expression levels of HA-tagged proteins by Western blot. The exosome-free supernatant was used to measure CD4-αCD3 expression level by immunoprecipitation-Western blot method. (**A**) Exo-Tat, IL16lamp2b and CD4-αCD3 expressed in AAV-Eliminator-infected Molt-4 cells. (**B**) Exo-Tat and IL16lamp2b were detected in the exosomes purified from the supernatant of AAV-Eliminator-infected Molt-4 cells. (**C**) CD4-αCD3 was detected in the exosome-free.

**Figure 7 microorganisms-12-01707-f007:**
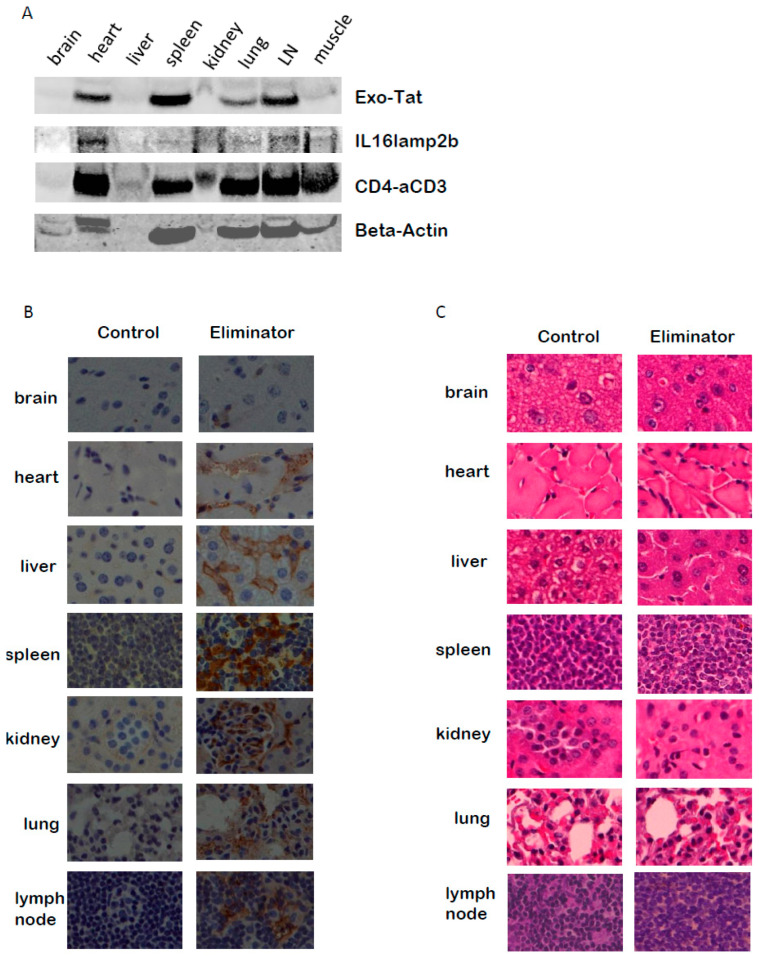
AAV delivered protein expression in vivo. AAV-DJ-GFP viruses (control) or AAV-Eliminator viruses (Eliminator) were injected intravenously into Balb/cJ mice via tail veins (1 × 10^12^ GC/mouse in 100 μL PBS). Thirty-one days post injection, the mice were euthanized with overdose isoflurane and various tissues were taken out for Western blot, immunohistochemistry (IHC) or H&E staining. (**A**) Western blot showing HA-tagged proteins expressed in various tissues. (**B**) IHC staining showing HA-tagged proteins express in various tissues (Brown color). (**C**) H&E staining showing AAV delivered target proteins do not change the structures of the tissues.

**Figure 8 microorganisms-12-01707-f008:**
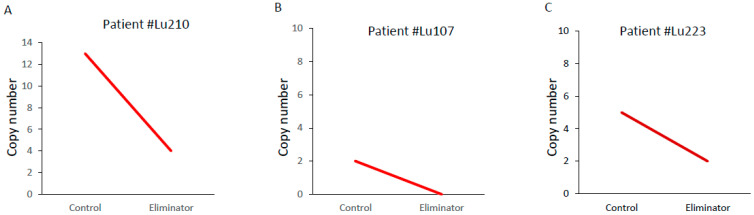
AAV-Eliminator reduces latent HIV-1 reservoir in T cells isolated from HIV-1-infected, cART-treated patients. One million Huh-7 cells were infected with control AAV-GFP (Control) or AAV-Eliminator (Eliminator) at MOI = 2 × 10^4^. Two days post infection, the cells were washed with PBS and cocultured with 4 million T cells isolated from HIV-1-infected patient PBMCs in RPMI 1640 medium supplemented with penicillin-streptomycin, L-glutamine, 0.1 nM IL-7, 1 μM tenofovir, 1 μM nevirapine, 1 μM emtricitabine. Four days later, T cells were collected and used for total DNA preparation. Three μL out of 120 μL DNA were used for each ddPCR reaction. Three replicates of ddPCR reactions were performed for each patient sample. (**A**) AAV-Eliminator reduces latent HIV-1 reservoir in T cells isolated from patient #Lu210. (**B**) AAV-Eliminator reduces latent HIV-1 reservoir in T cells isolated from patient #Lu107. (**C**) AAV-Eliminator reduces latent HIV-1 reservoir in T cells isolated from patient #Lu223.

## Data Availability

The original contributions presented in the study are included in the article, further inquiries can be directed to the corresponding author.

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
