# Peer review of "Adeno-Associated Virus (AAV)-Delivered Exosomal TAT and BiTE Molecule CD4-αCD3 Facilitate the Elimination of CD4 T Cells Harboring Latent HIV-1"

_microorganisms, 2024, doi:10.3390/microorganisms12081707_

Round 1

Reviewer 1 Report

Comments and Suggestions for Authors

A summary (one short paragraph) outlining the paper's aim, main contributions, and strengths.

The manuscript titled AAV-delivered exosomal Tat and BiTE molecule CD4-á¾³CD3 facilitates the elimination of CD4 T cells harboring latent HIV-1. The platform wherein they use Exo Tat to stimulate latent virus from the quiescent CD4 T cells was intended to be presented. This is achieved by using the particular binding domain CD4 in interleukin 16 (IL16) attached to the N terminus of the exosome surface protein, lysosome-associated membrane protein 2 variant B. (Lamp2B).  Prior research by this group demonstrated that exosomal Tat (Exo-Tat) may reactivate latent virus from primary CD4 T cells in PWH. This publication demonstrates that adding a retargeting protein with affinities for both CD3 and gp120 causes HIV-1-infected cells to perish. Exo-Tat, IL16lamp2b, and Cd4-alfaCD3 may be delivered via a viral platform to eradicate HIV-1 reservoirs in PWH primary lymphoid cells. They also discuss the development and validation of this immune agent. This study has a great impact on the field of HIV-1 cure. If the authors can be considered to increase the number of people to test this product.

  • General concept comments

The current study has developed a new strategy for eliminating the virus from resting CD4 T cells using both the immunological and potential platform to potentially deliver Exo Tat, IL16lamp2b, and CD4-á¾³CD3 to potentially eliminate the reservoirs from PWH. This is the significant strength of the paper since it targets both. Using both in vitro and in vivo studies has also strengthened the validity of these results. The weakness of this study is the number of PWH where this method was tested. The numbers are too low to conclude with the results.

1.     Title: the title of the manuscript can be rephrased; instead of AAV-delivered you can write them in full instead of abbreviation.

2.     Introduction: Line 46 “Five individuals have been cured of HIV-1 infection” May authors provide information on how these five individuals and how the methods cannot be used further.

§  Line 52: Explain the several experimental latency reversal strategies and why they failed. How each strategy is different from each other.

§  Your new strategy of activating latency will be shock and kill or Block and lock?

§  The authors can also provide more background information or rationale for using the AAV delivery system. Why do you think it is the best compared to previously used techniques?

3.     Where did you get the HEK293T and Huh-7 cells? Were they purchased or donated

4.     Describe the MOLT-4/CCR5 cell

5.     HIV-p24 Elisa

§  Line 147-149 that statement can moved to section 2.13 instead

6.     Under study subject section

§  Authors need to mention the number of participants that were involved in the study. Instead of saying HIV-1 infected individual, they can be addressed as people living with HIV-1 (PLWH)

§  It will be advisable that the author mention whether the treatment was initiated early in the infection during the chronic stage.

§  Since the participants were on treatment: May you provide the list that were on?

§  What was the control group used in this experiment?

7.     Production of AVV viruses

§  During the production of AAV viruses it is not clear whether you harvested the cells or supernatants. If cells were harvested may the authors why the viruses would be in the cell not in the supernatant

8.     Figure legends are too long and most of the information in the figure legends can be moved to material and methods. E.g. Figure 5 Lines 368-line 384 must be under material and methods and explain what the x and y axes represent. You can revise all your figure legends.

9.     Figure 8: The axis is written as a copy number. What are you referring to?

Additionally, authors may also provide the originals of cell lines that have been used in this study and why are they suitable for the study.

Figure legends need to be revised and the explanation of figures is insufficient

Reviewer 2 Report

Comments and Suggestions for Authors

In this manuscript, the authors explore a novel approach to potentially curing HIV-1 by targeting latent reservoirs of the virus in infected individuals. They developed an HIV-1 eradication platform using engineered exosomal Tat (Exo-Tat) to reactivate latent HIV-1 in CD4+ T cells. This approach was combined with a bispecific T-cell engager (BiTE) molecule, CD4-αCD3, which aids the immune system in targeting and eliminating these reactivated, HIV-infected cells. The study presents a promising strategy for targeting and reducing latent HIV-1 reservoirs, offering a potential pathway towards an HIV cure. While the manuscript is promising, it could be strengthened by addressing some minor issues, such as adding subtitles under each sub-figure in Figures 1, 2, 4, and 5 to help readers compare the data more easily.

Author Response

Please see the attached file for our response to the reviewer's comment. Thank you very much.

Round 2

Reviewer 1 Report

Comments and Suggestions for Authors

The authors has adequately addressed the comments that I have raised. I satisfied with the responses.